# Defining the hidden burden of disease in rural communities in Bangladesh, Cambodia and Thailand: a cross-sectional household health survey protocol

Meiwen Zhang,[1,2] Nan Shwe Nwe Htun,[1] Shayla Islam,[3] Aninda Sen [ID] ,[3] Akramul Islam,[3] Amit Kumer Neogi,[3] Rupam Tripura,[1] Lek Dysoley,[4] Carlo Perrone,[1] Rusheng Chew [ID] ,[1,2,5] Elizabeth M Batty,[1,2] Watcharintorn Thongpiam,[1] Jantana Wongsantichon,[1] Chonticha Menggred,[1] Sazid Ibna Zaman,[1,2] Naomi Waithira,[1,2] Stuart Blacksell,[1,2] Marco Liverani,[6,7,8] Sue Lee [ID] ,[1,2,9] Richard James Maude [ID] ,[1,2,10] Nicholas P J Day,[1,2] Yoel Lubell,[1,2] Thomas Julian Peto [ID] [1,2]

For numbered affiliations see end of article.

**Correspondence to**
Dr Meiwen Zhang;
meiwen@tropmedres.ac

## ABSTRACT

**Introduction** In low-income and middle-income countries in Southeast Asia, the burden of diseases among rural population remains poorly understood, posing a challenge for effective healthcare prioritisation and resource allocation. Addressing this knowledge gap, the South and Southeast Asia Community-based Trials Network (SEACTN) will undertake a survey that aims to determine the prevalence of a wide range of non-communicable and communicable diseases, as one of the key initiatives of its first project—the Rural Febrile Illness project (RFI). This survey, alongside other RFI studies that explore fever aetiology, leading causes of mortality, and establishing village and health facility maps and profiles, will provide an updated epidemiological background of the rural areas where the network is operational.

**Methods and analysis** During 2022–2023, a cross-sectional household survey will be conducted across three SEACTN sites in Bangladesh, Cambodia and Thailand. Using a two-stage cluster-sampling approach, we will employ a probability-proportional-to-size sample method for village, and a simple random sample for household, selection, enrolling all members from the selected households. Approximately 1500 participants will be enrolled per country. Participants will undergo questionnaire interview, physical examination and haemoglobin point-of-care testing. Blood samples will be collected and sent to central laboratories to test for chronic and acute infections, and biomarkers associated with cardiovascular disease, and diabetes. Prevalences will be presented as an overall estimate by country, and stratified and compared across sites and participants' sociodemographic characteristics. Associations between disease status, risk factors and other characteristics will be explored.

**Ethics and dissemination** This study protocol has been approved by the Oxford Tropical Research Ethics Committee, National Research Ethics Committee of

## STRENGTHS AND LIMITATIONS OF THIS STUDY

⇒ The study will use diverse methods (interviews, physical examinations and laboratory tests) to gather extensive data on the prevalence, risk factors, healthcare utilisation patterns of both communicable and non-communicable diseases.

⇒ The use of two-stage cluster sampling enables the sample to be representative of each study area.

⇒ As an integrated component of the Rural Febrile Illness project, the findings will be synergised with outcomes of concurrent studies in these specified regions, offering a multidimensional overview of health and healthcare provision.

⇒ The findings are specific to the selected rural areas and should be interpreted with caution when considering broader implications for rural populations in the study countries.

Bangladesh Medical Research Council, the Cambodian National Ethics Committee for Health Research, the Chiang Rai Provincial Public Health Research Ethical Committee. The results will be disseminated via the local health authorities and partners, peer-reviewed journals and conference presentations.

**Trial registration number** NCT05389540.

## INTRODUCTION

Life expectancy in Southeast Asia increased from 63 to 71 years between 2000 and 2019.[1] These positive changes can be attributed, in part, to advancements in the region's health systems, which have addressed traditional high-burden diseases, such as infectious diseases, maternal and neonatal health, and

under 5 mortality.[2] Despite these improvements, the burden of traditionally significant diseases persists, particularly in rural areas. Furthermore, there is a transition in disease epidemiology due to rapidly changing environments, growing economies and ageing populations.[3] The shift is marked by a rising burden of non-communicable diseases and injuries, along with emerging infectious diseases (eg, COVID-19 and chikungunya).[4 5]

The knowledge of current disease epidemiology in the region, however, is limited, constraining health systems' ability to identify healthcare priorities and direct future resource allocation.[6] While the Global Burden of Disease study provides insights into disease burdens across countries and diseases, the accuracy of its results is impeded by the scarcity of recent epidemiological data.[6] Given the limited coverage of disease reporting and surveillance systems in the region, research data are pivotal in bridging this information gap. Such data not only offer up-to-date information for modelling studies but are also critical for optimising resource allocation by including factors such as disease severity, the magnitude of the population affected and equity.[7]

Moreover, the marked systemic disparities in healthcare between rural and urban areas in the region highlight the need for rural-specific epidemiological data.[3] Despite 50%–85% of the region's population residing in rural areas, this large population group remains relatively understudied compared with countrywide estimates.[8–13] In this context, cross-sectional surveys are an effective research method to provide timely estimates of disease prevalence in rural communities, through cost-effective approaches combining questionnaire interviews, clinical examinations and laboratory tests.

The newly formed South and Southeast Asia Community-based Trials Network (SEACTN) aims to establish a network of community-based healthcare providers and facilities capable of implementing interventions designed to triage, diagnose and treat patients within rural communities across five South and Southeast Asian countries (Bangladesh, Cambodia, Laos, Myanmar and Thailand).[14] The first project is the Rural Febrile Illness Project (RFI), dedicated to delineating the epidemiological baseline of febrile illness in remote and underserved regions, where febrile illness and access to healthcare pose significant health challenges.[14] RFI encompasses diverse initiatives designed to gain a multifaceted understanding of the health dynamics within these communities to effectively facilitate the identification of interventions for future studies across SEACTN. Recognising the gaps in understanding of disease prevalence, a cross-sectional household health survey has been planned as one of the key initiatives of the RFI.

This survey aims to define the prevalence of a broad spectrum of communicable and non-communicable causes of health conditions in areas where the network operates. Alongside the survey, other key initiatives of RFI include a fever aetiology study to determine the incidence, causes and outcomes of febrile illness; a verbal autopsy study to identify common causes of mortality and the circumstances surrounding death; and a village and health facility mapping study creating a detailed profile of the study villages, estimating travel time to health facilities and identifying health service provision gaps. Each initiative holds significant value as an independent study, and their results will be complementary providing a thorough comprehension of healthcare needs and enabling more equitable resource allocation.

## METHODS AND ANALYSIS

The SEACTN household health survey is a community-based cross-sectional survey aiming to provide an overview of the burden of disease in selected rural areas where the network is operational. The first participant was enrolled on 3 October 2022, and data collection is expected to continue until December 2023.

### Primary objectives and outcomes

1. To determine the prevalences of selected diseases and exposure to locally prevalent or important pathogens, including, but not limited to:
   a. IgG against selected common pathogens causing fever.
   b. Hepatitis B and C.
      i. Hepatitis B virus (HBV) surface antigen.
      ii. IgG against Hepatitis C virus (HCV) and presence of HCV.
   c. Selected non-communicable diseases (eg, diabetes, raised blood cholesterol, hypertension, stroke) according to self-reported disease history, and/or physical examinations and/or laboratory tests.
   d. Self-reported illness or injury in the past 30 days.
      i. Any acute conditions (eg, fever, persistent cough, watery diarrhoea).
      ii. Injury or death caused by an accident.
2. To determine the point prevalence of different self-perceived health statuses.
3. To determine the prevalences of major risk factors for common non-communicable diseases (eg, smoking, alcohol consumption, overweight).

### Study population and setting

All SEACTN operational areas were selected by partner organisations as they represent poor, remote and rural communities with limited access to formal healthcare.[14] This study will be conducted in three areas, consisting of 391 villages in Bangladesh, Cambodia and Thailand (figure 1). The other sites will not take part of the study due to operational constraints.

A two-stage cluster sampling will be used to obtain a representative sample from each study site. In the first stage, 75 villages will be selected using the probability-proportional-to-size sample method.[15–17] For the second stage, within each village, a simple random sample of 5–7 households (adjusted to the average household size) will

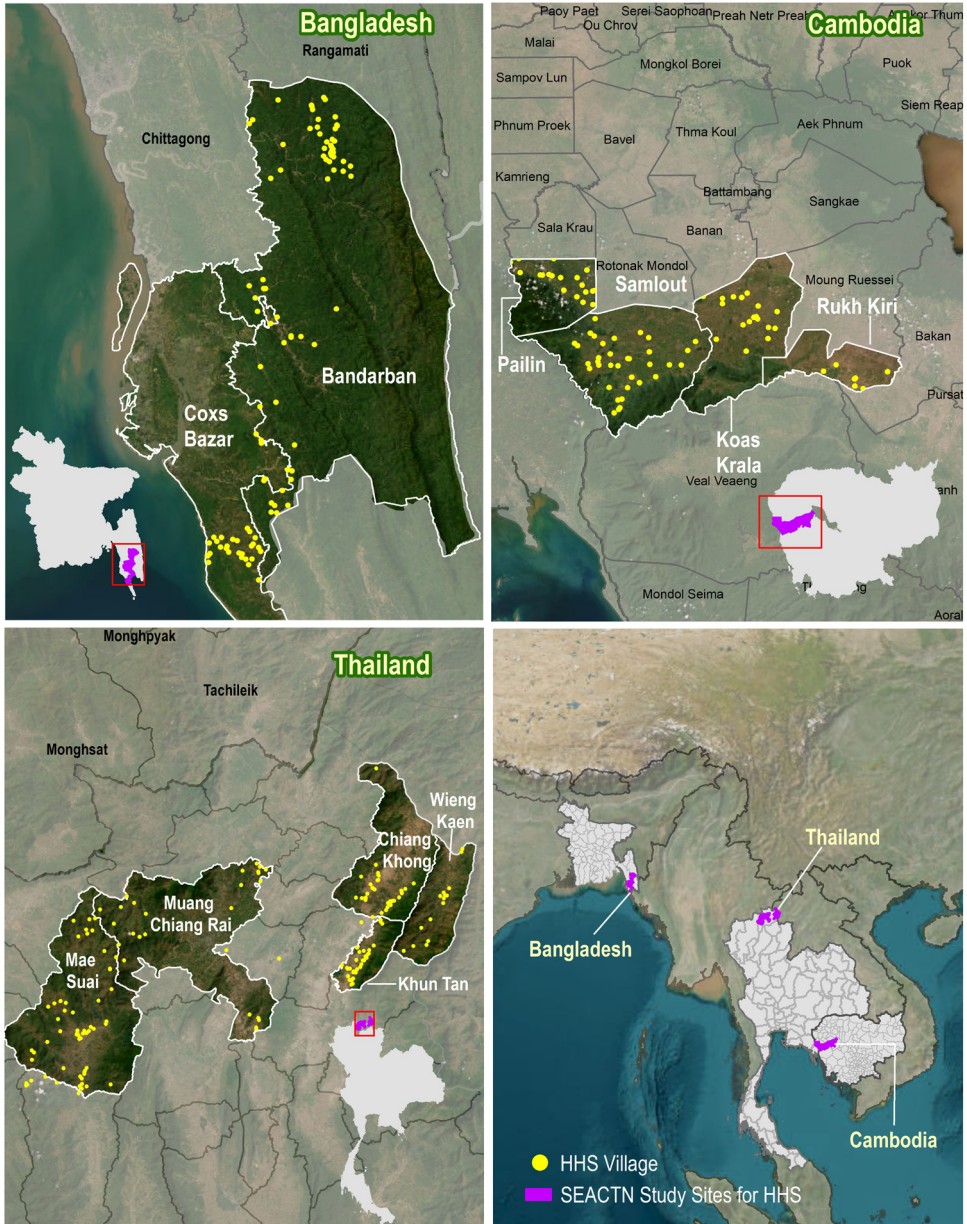

**Figure 1** The study sites and population of the cross-sectional household health survey conducted by the South and Southeast Asia Community-based Trials Network (SEACTN). The Bangladesh site includes 130 villages from Cox's Bazar and Bandarban districts; the Cambodia site includes 82 villages from Battambang and Pailin provinces; and the Thailand site includes 179 villages located in Chiang Rai province. HHS, The Household Health Survey.

be selected using computer-generated random sampling from the village household list.

### Inclusion and exclusion criteria

For each selected household, all usual household members or visitors who stayed overnight in the households before the survey are eligible to be enrolled.[18]

### Sample size

This study is adequately powered to determine the prevalences of all key indicators within each age group, with a design effect of 1.5 and a range of 5%–10% degrees of precision, while considering a maximum of 1500 participants per study country (table 1). To account for a 10%

non-response rate, assuming an average household size of 4.5 persons, and enrolling all household members, the final sample size was adjusted to 1667 participants per country.[19–21]

The minimal sample size required was calculated based on the estimated prevalences of key indicators overall and within each population age group (table 1). Prevalence estimates, population proportions within each age group and average household size were derived from previously conducted national health surveys and published studies, preferably from the study country, or from countries with a similar epidemiological context.[19–29] When the estimated prevalence of an

**Table 1** Required sample size tested against the estimated prevalence of key indicators of the highest prevalence, overall and by age groups of interests

| | Expected number of participants | Indicator (estimated prevalence) | Required sample size* |
|---|---|---|---|
| Overall | 1500 | Overweight (20%) | 369 |
| Age groups (years) | | | |
| ≥15 | 1050 | Current drinker (55%) | 570 |
| <5 | 150 | Anaemia (30%) | 122 |
| 15–49 | 750 | Injury or death caused by recent accident (20%) | 414 |
| ≥50 | 300 | Hypertension (20%) | 257 |

$$n = DEFF \times \frac{Np(1-p)}{\frac{d^2}{1.96^2}(N-1)+p(1-p)}$$

n: sample size; DEFF: design effect. Design effect of 1.5 is used to account for a slightly expected increase in variance due to clustering; N: population size; p: expected prevalence of key indicators; d: degree of precision.
*The formula used for calculating sample size is shown above. Assessing the sufficiency of the sample size for the key indicators of the highest prevalences ensures sufficient statistical power for indicators with lower prevalence. A degree of precision of 5% is applied for the sample size calculation overall and across age groups, except for the under 5 years and the 50 years or older groups, for which degrees of precision of 10% and 6% are applied, respectively.

indicator varied across study country, the highest prevalence estimates were applied.

## Study procedures

Standard operating procedures (SOPs) were developed for field procedures, sample collection and transportation. Training materials were developed and training sessions were delivered to all the study staff directly, or using a train-the-trainer model where field supervisors and managers will be trained first and then subsequently train other field staff. All research staff are experienced in community-based research or interventions. Experienced phlebotomists were recruited and trained in specimen collection and processing in the field. A monitoring and evaluation plan was devised to maximise the research quality.

## Patient and public involvement

The potential challenges for the study include comprehensibility of the questionnaires, and willingness of selected households to participate. We have proactively addressed these concerns by involving the target population in the questionnaire validation process (section: Questionnaire interviews), and planning community

mobilisation events before the survey, tailored to the local context.

Collaborators will identify contact points from local stakeholders and communities, such as village leaders, local health centre staff or community health workers, and determine the most suitable methods for establishing contact. Main events may involve community engagement meetings to explain the study purpose, organisation and procedures, and the plan for dissemination of study results. Prior to the survey day for each village, the study team will meet, or call if physical meetings are not feasible, the household head or members of the selected households. With the support of the key contact persons, this interaction will create an opportunity to address questions or concerns, enabling the survey teams to work with each community and selected household to plan an effective data collection schedule.

## Survey schedule

On the scheduled survey day for each village, stations will be set up at a convenient location within or near the village. Household members will arrive at the survey location following their appointment to complete the procedures (figure 2). If household members do not attend the data collection at the appointment time, they will be contacted and encouraged to join the survey at another time during the day, while the research team is in the village.

## Informed consent

Participants must sign the informed consent form before any study-specific procedures are performed. The informed consent will be available in the official local language. It will be presented to prospective participants by trained study staff detailing the study procedures and implications of study participation. It will be made explicit that participation is voluntary and participants are free to withdraw from the study at any time. If necessary, the informed consent process will be interpreted to dialect when participants do not speak the official local language.

Adequate time will be given to study participants to consider the information and ask questions. Written consent will then be obtained from the participants, or the caretakers of participants, in compliance with local legal age requirements, using the participant's dated signature or thumbprint (if unable to write) and dated signature of a person who presented and obtained the informed consent. A copy of the signed informed consent document will be given to the participant. Children aged as dictated by local legislation and regulations will be required to sign the written informed assent form in addition to their parent or guardian signing a consent form.

## Questionnaire interviews

The questionnaire consists mostly of questions adapted from well-validated and widely implemented tools (online supplemental appendix A). This approach enables direct

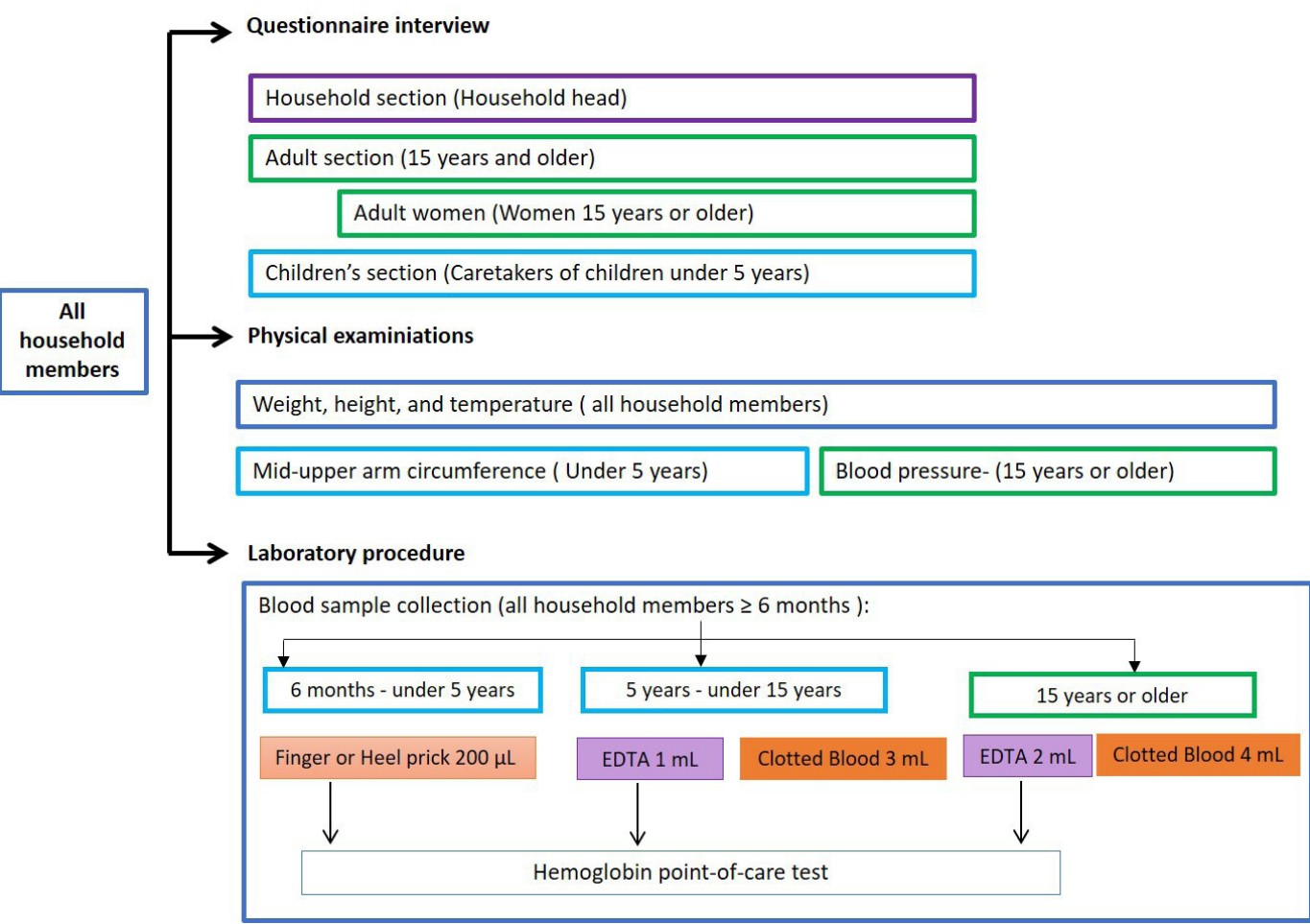

**Figure 2** Questionnaire interviews, physical examinations and laboratory procedures administrated to study participants according to age and sex.

comparison of results with previously conducted surveys by incorporating questions from the Demographic and Health Surveys, WHO STEPWise surveys for Non-communicable Disease Risk Factors (STEP) and the World Health Survey.[30–32]

The questionnaire underwent adjustments and validation through pretests using behaviour coding and participant debriefing.[33 34] During mock interviews, the observer of each interview pair recorded the interaction problems for each question, such as 'major change in wording', 'question reading interrupted by the respondent' or 'additional prompt is provided'. All interviewers, interviewees and the observers participated in the debriefing. The pretest was conducted initially among peer researchers in English. Subsequently, the adjusted version was translated into Thai and tested among the target population in Thailand. This ensured the questionnaire's suitability, comprehensibility and appropriate length for administration. The final questionnaire was then translated into the official languages of each study country (Bangla, Khmer and Thai) by professional translators or dedicated staff from partner organisations, and validated by the local research teams. When necessary, during face-to-face interviews, questions will be interpreted to dialect for participants who do not speak the official local language.

The questionnaire consists of four sections answered by household members based on their sex and age (figure 2, top).

▶ The household section is completed by the household head, providing information on household characteristics and a list of all household members along with any illnesses in the past 4 weeks and health seeking behaviour.

▶ The adult section is for household members aged 15 years or older and includes questions on sociodemographic characteristics (eg, age, sex, education, marital status and occupation), disease history (eg, tuberculosis, diabetes, cardiovascular disease), self-perceived health status, health concerns, substance use (eg, tobacco and alcohol) and other disease risk factors.

▶ The women's health section is for female household members aged 15 years or older and includes questions on contraception use, utilisation of antenatal care and delivery services.

► The children's health section is answered by caretakers of children under 5 years, providing information on each child's vaccination and breastfeeding history.

### Physical examination

All household members will undergo weight, height and tympanic temperature measurements. Children under 5 years will also have their mid-upper arm circumference measured, and adult household members (≥15 years) will undergo blood pressure measurement (figure 2, middle). Weight will be measured with a digital scale, preferably with a taring function allowing a child's weight to be measured while being held by an adult. Height measurements will be carried out with measuring scales/boards. Children younger than 24 months will be measured lying down on the board, while standing height will be measured for older children. Tympanic temperature will be measured with infrared thermometers.

All measures will be taken by experienced and trained staff. Consistency and comparability will be ensured by using the same validated devices previously used in studies in similar settings, whenever possible, across study sites.

### Blood sample collection and point-of-care haemoglobin test

Blood samples will be collected from participants aged 6 months or older (figure 2, bottom). For participants aged 6 months to under 5 years, four dried blood spots (DBS) will be collected via finger or heel prick. Participants aged 5 years or older will provide venous blood samples, with 4 mL collected from those aged 5 to under 15 years, and 6 mL from participants aged 15 years or older (figure 2, bottom). After blood sample collection, appropriate samples will be tested for haemoglobin using HemoCue Hb 301 system (HemoCueAB, Ängelholm, Sweden) to detect anaemia.

The venous blood samples will be temporarily stored in cool boxes between 4°C and 8°C, for a maximum of 24 hours before centrifugation in the field laboratories. Aliquots of whole blood, serum, plasma and packed red blood cells of these samples will be stored at a minimum of −20°C or below without freeze-thaw until analysis. DBS will be air-dried in the laboratory for at least 6 hours, after which they will be placed in a plastic zip-lock bag and stored at room temperature or at 4°C or below.

### Information, consultation and referral

On completion of the field procedures, individual participants will receive the results of their haemoglobin test and physical examinations. Participants with abnormal results will be referred to a study nurse who will provide recommendations and, if the participant wishes, further referrals in accordance with local guidelines.

### Non-responders

Household members who fail to attend the survey (who are absent, refuse participation or for other reasons) will have their demographic characteristics (eg, sex, age) summarised from the information obtained from the household head in the household section of the questionnaire, to determine whether systematic differences exist between responders and non-responders.

### Analysis of blood samples

Sample aliquots and DBS will be shipped to the central laboratories in Bangkok, Thailand where they will be analysed. All laboratory activities will be performed by experienced and trained staff. All laboratory procedures will be performed using validated SOPs. Multiplex serology tests based on Luminex xMAP Intelliflex System will be developed and validated for measures of IgG and IgM of common pathogens causing fever; all other methods have already been published in peer-reviewed journals.

Malaria quantitative PCR (qPCR) will be performed for all participants to identify individuals with malaria parasites. The PCR methods typically have a detection level in the range of 100–1000 parasites per millilitre.[35] Genus-specific qPCR targeting Plasmodium *18S rRNA* genes will first be performed to screen for positive individuals, and then species-specific assays will be performed to differentiate the parasite species (*Plasmodium falciparum, Plasmodium vivax, Plasmodium malariae, Plasmodium ovale* and *Plasmodium knowlesi*).[36–39]

Serology tests for selected common pathogens causing fever including IgG and IgM for Dengue, Chikungunya, Japanese encephalitis, Zika, SARS-CoV-2, *P. vivax, P. falciparum, Orientia tsutsugamushi, Rickettsia* spp and *Leptospira* spp will be performed for all participants.

Tests for hepatitis B and C: all participants will be tested for HBV surface antigen; participants 15 years or older will also be tested for IgG for HCV, and the positive samples will undergo HCV PCR.

Non-communicable disease-related markers: participants 15 years or older will be tested for Haemoglobin A1c, total cholesterol and high-density lipoprotein cholesterol.

Additionally, the remaining samples will be tested for exposure to, or presence of, pathogens associated with selected diseases that are regionally or locally prevalent or important. The final list of diseases, pathogens and sample selection will be adjusted based on findings from other SEACTN studies exploring diagnostics and prognostics in febrile patients, and other available evidence.

### Results feedback

Individual-level data collected from physical examinations and haemoglobin tests will be provided on the same day once all field procedures are concluded for an individual participant (section: Study procedure—Information, consultation and referral). Additionally, participants will be thoroughly informed about the implications of the laboratory tests to be performed during the consent process. They will be presented with the option to receive the results of hepatitis B and C tests. If they wish to receive them, the results, once available, will be placed in a sealed envelope and will then be delivered by the local health providers or village leaders to the participants. It

is anticipated that there will be a delay of approximately 6 months to 1 year from sample collection to results feedback. The time frame encompasses the completion of data collection at the entire site, transportation of samples to Bangkok, sample management, analysis and the subsequent return of results. The decision on which results to provide to individual participants is made in consultation with the site research teams and local clinicians, considering factors such as their clinical relevance, the potential harm that could result from a lack of consultation on results distribution and the accessibility of treatment.

## Planned analysis

All data obtained through questionnaire interviews, physical examinations and haemoglobin point-of-care tests will be collected on tablets using an electronic data collection platform—Open Data Kit.[40] The electronic case report form has built-in validation rules to identify missing or potentially incorrect data. Stringent checks are applied to variables used to estimate the study outcomes, such as recent fever history, diabetes history and weight. Survey interviewers cannot progress to the next question if these data are missing or outside the defined ranges. Throughout the survey, data will be continuously updated and monitored for quality. Data queries and quality reports will be generated every two to 4 weeks for data verification and correction, and to identify areas needing additional training and support, ensuring data integrity.

For descriptive summaries, means and SDs or medians with IQR for continuous variables, and proportions for categorical variables will be calculated. Outcomes will be presented as an overall estimate by country, and with 95% CIs, stratified and compared across sites and participants' sociodemographic characteristics using appropriate tests, such as the Student's t-test, Mann-Whitney U tests or $\chi^2$ tests. Associations between disease status, risk factors, self-perceived health status and other characteristics will be explored through univariate and multivariate analyses, and measures of effect will be reported with 95% CIs. Sampling weights will be accounted for, as needed, in the analysis.

## DISCUSSION

This prevalence survey conducted across three rural SEACTN study sites in Bangladesh, Cambodia and Thailand aims to provide a comprehensive epidemiological description of the study areas. Employing a multifaceted approach, including questionnaire interviews, laboratory tests and physical examinations, the study will yield extensive data on the prevalence, risk factors, healthcare utilisation patterns of both communicable and non-communicable diseases. The findings will contribute to improved estimates of the burden of disease, and deepen the comprehension of findings from other RFI initiatives, offering a thorough overview of health and healthcare provision in these areas. The survey will be invaluable for

setting healthcare priorities and directing resource allocation for health system development.

As a core component of RFI, this survey, alongside other initiatives, adopts an integrated approach to address health challenges more effectively, particularly in the context of the 'double burden of disease', characterised by the rising burden of non-communicable diseases, alongside ongoing challenges posed by infectious diseases and suboptimal maternal and child health.[2] This survey will encompass various dimensions of health, allowing for further exploration of associations among disease occurrence, risk factors and self-perceived health. Key data from the survey, such as household wealth status, and community members' perception of their health and health concerns, will contextualise the findings from the other three RFI studies, and provide insights into potential health disparities and inequalities. By integrating disease prevalence with fever aetiology and causes of death, a multilayered understanding of health dynamics in the study area could be attained. Additionally, knowledge from the survey will play a vital role for estimating incidence. This involves assimilating data on proportion of the population with fever who visited various healthcare providers from the survey, with the number of fever patients recruited from the village health workers or health facilities, captured from the fever aetiology studies.

The study will also bridge the gap in community-based seroprevalence estimates for common pathogens causing fever, minimising biases related to health-seeking behaviour in facility-based studies. The data can also to be used for population-based incidence estimates.[41 42] Furthermore, the updated prevalences of hepatitis B and C can be instrumental in evaluating the coverage and effectiveness of HBV vaccination programmes, and inform HCV screening and treatment strategies in the region in the era of direct-acting antiviral agents.[43 44] This study also extends its scope to gather insights into the health perspectives and concerns of community members, facilitating the alignment of future interventions with the needs and expectations of the population. Specifically, for self-perceived status, while existing studies primarily focus on the elderly population, this study encompasses a broader age range.[45–48] This can inform tailored interventions to improve the quality of life for a more extensive population.[49–51]

This study encourages regional collaboration by applying consistent research methods and seeking representative samples across study sites, allowing effective cross-country comparisons and fostering collaborative approaches to address shared health challenges.[3]

The findings of this study are limited in their generalisability and should be interpreted with caution when considering broader implications for rural populations in the study countries. However, the study was specifically designed for and conducted within the selected rural areas of Bangladesh, Cambodia and Thailand, where SEACTN is operational. This focused approach allows for insights into health challenges and opportunities within

these areas. The cross-sectional design and observational nature of the study is at risk of, such as temporal bias, selection bias and participation bias. However, to minimise the bias as much as possible, we designed the survey with appropriate evaluation methods, and adequate training and supporting material will be well prepared for the field work.

This planned household prevalence survey will significantly contribute to disease prevention and control in the region by offering a comprehensive understanding of health conditions of the rural populations. Complementing the results from other RFI initiatives, it will serve as a foundation for evidence-based interventions, shaping future research and disease control priorities. The study findings will not only serve the next phase of SEACTN but also extend to the broader context, ultimately leading to improved health outcomes in rural communities.

## ETHICS AND DISSEMINATION
This study protocol was approved by the Oxford Tropical Research Ethics Committee (OxTREC Ref: 6-22), National Research Ethics Committee of Bangladesh Medical Research Council (BMRC/NREC/2022-2025/69), the Cambodian National Ethics Committee for Health Research (23 December 2022 NECHR), the Chiang Rai Provincial Public Health Research Ethical Committee (CRPPHO 75/2565).

### Safety considerations
Participants may experience slight discomfort during the blood draw, however, our experienced clinical staff performing the phlebotomy will take special attention to minimise this discomfort. While some interview questions may be personal, all data will be handled confidentially and linked only to study IDs. Participants will also be informed of their right to skip questions or end the interview if they feel uncomfortable.

### Data deposition and curation
The database and all electronic data will be stored on secure servers that are backed up daily, with weekly off-site storage. Paper records, if they exist, will be kept in secure storage such as locked cabinets; if necessary, the record will be scanned and stored electronically. Study data will be archived in accordance with Mahidol Oxford Tropical Medicine Research Unit (MORU) SOPs. Electronic data will be deidentified and preserved indefinitely. Paper records will be preserved for 5 years after study completion. With participant's consent, electronic data can be shared according to the terms defined in the MORU data sharing policy with other researchers for future use.[52]

### Dissemination
The results of the study will be summarised in plain language in both English and the official local language. These summaries will be disseminated via the local health authorities and partners. The findings will also be disseminated internationally through conference presentations and peer-reviewed academic journal publications, aligning with Wellcome Trust policy and guidelines.

**Author affiliations**
[1]Mahidol Oxford Tropical Medicine Research Unit, Bangkok, Thailand
[2]Centre for Tropical Medicine and Global Health, Nuffield Department of Medicine, University of Oxford, Oxford, UK
[3]Communicable Diseases Program, BRAC, Dhaka, Bangladesh
[4]Cambodian National Malaria Control Program, Phnom Penh, Cambodia
[5]Faculty of Medicine, University of Queensland, Brisbane, Queensland, Australia
[6]Department of Global Health and Development, London School of Hygiene & Tropical Medicine, London, UK
[7]Faculty of Public Health, Mahidol University, Bangkok, Thailand
[8]School of Tropical Medicine and Global Health, Nagasaki University, Nagasaki, Japan
[9]Department of Infectious Diseases, Monash University and Alfred Hospital, Melbourne, Victoria, Australia
[10]The Open University, Milton Keynes, UK

**Acknowledgements** The authors are grateful for the support from the local authorities in the participating countries. We acknowledge all staff from our partners and collaborators for their devoted contributions to the survey.

**Contributors** MZ, NSNH and TJP contributed to conception and design of the study, and drafted the manuscript. SI, AS, MI, AKN, RT, LD, CP, RC, EMB, WT, JW, SDB and SL contributed to the design of the study and provided critical review of the manuscript. NW and CM contributed to the design of the study, data collection platform development and provided critical review of the manuscript. SIZ contributed to the design of the study, provided the study sites' map and critical review of the manuscript. ML, RJM, NPJD and YL contributed to the conception of the study and provided critical review of the manuscript.

**Funding** This research was funded in whole, or in part, by the Wellcome Trust [220211]. For the purpose of Open Access, the author has applied a CC BY public copyright licence to any Author Accepted Manuscript version arising from this submission.

**Map disclaimer** The inclusion of any map (including the depiction of any boundaries therein), or of any geographic or locational reference, does not imply the expression of any opinion whatsoever on the part of BMJ concerning the legal status of any country, territory, jurisdiction or area or of its authorities. Any such expression remains solely that of the relevant source and is not endorsed by BMJ. Maps are provided without any warranty of any kind, either express or implied.

**Competing interests** None declared.

**Patient and public involvement** Patients and/or the public were involved in the design, or conduct, or reporting, or dissemination plans of this research. Refer to the Methods section for further details.

**Patient consent for publication** Not applicable.

**Provenance and peer review** Not commissioned; externally peer reviewed.

**ORCID iDs**
Aninda Sen http://orcid.org/0009-0002-2243-4566

Rusheng Chew http://orcid.org/0000-0002-8992-4474
Sue Lee http://orcid.org/0000-0002-8409-4248
Richard James Maude http://orcid.org/0000-0002-5355-0562
Thomas Julian Peto http://orcid.org/0000-0003-3197-9891

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
