## [Reviewer comments · BMJ Open]

ARTICLE DETAILS

TITLE (PROVISIONAL)	DEFINING THE HIDDEN BURDEN OF DISEASE IN RURAL COMMUNITIES IN BANGLADESH, CAMBODIA AND THAILAND: A CROSS-SECTIONAL HOUSEHOLD HEALTH SURVEY PROTOCOL
AUTHORS	Zhang, Meiwen; Htun, Nan Shwe Nwe; Islam, Shayla; Sen, Aninda; Islam, Akramul; Neogi, Amit Kumer; Tripura, Rupam; Dysoley, Lek; Perrone, Carlo; Chew, Rusheng; Batty, Elizabeth M.; Thongpiam, Watcharintorn; Wongsantichon, Jantana; Menggred, Chonticha; Zaman, Sazid Ibna; Waithira, Naomi; Blacksell, Stuart; Liverani, Marco; Lee, Sue; Maude, Richard; Day, Nicholas; Lubell, Yoel; Peto, Thomas

VERSION 1 – REVIEW

REVIEWER	Khanal, Mahesh Bangladesh Institute of Health Sciences
REVIEW RETURNED	02-Dec-2023

GENERAL COMMENTS	I want to congratulate the authors for being able to bring the manuscript to this level. This manuscript is based on the ongoing project in rural communities across five South and Southeast Asian countries (Bangladesh, Cambodia, Laos, Myanmar, and Thailand). The cross-sectional study will determine the prevalence of selected non-communicable diseases and their risk factors, infectious diseases, and injuries. Data collection will be completed in Dec. 2023. The manuscript is well written. I only have minor revisions to be corrected before publication. Abstract:  • The objective is not clear. Is it the Rural febrile Illness project or different? Please clarify the objectives Method and analysis  • Please specify the non-communicable diseases, if it is already selected. • What is the plan for the family members not in the selected house during the data collection day? Discussion:  • Are there any possibilities of limitation apart from the selected location?
--

REVIEWER	von Fricken, Michael George Mason University
REVIEW RETURNED	15-Jan-2024

GENERAL COMMENTS	In your intro you cite the 2019 global burden of disease study when describing the absence of survey data for Cambodia,
---

	Myanmar, and Laos, but that doesn't align with your study sites – where you are looking at Bangladesh, Cambodia, and Thailand. Please update, remove, or provide correct context. If you are using GBD I believe there are more recent publications of GBD study as well. For your power analysis, shouldn't your target sample size been set on the underlying burden of disease by country? I would suspect the prevalence of some of these diseases would be very different in Thailand vs Bangladesh vs Cambodia – is there any risk of study being underpowered in any location by adopting a one sample size (~1600 per country) fits all approach? Please provide more detail on your PCR targets for malaria in your analysis of blood samples section. Same for serology tests – are you running ELISAs? IFA? Western blot? MagPIX? A little more granularity here would help flesh out this published protocol. Within your results feedback step – what type of delays are you anticipating between collection event and test results for all HBV, HCV, etc. please include an estimate if available I have no comment on the survey instrument, especially since this study is approaching completion at the time of this review.
--	---

VERSION 1 – AUTHOR RESPONSE

Reviewer 1

Dr. Mahesh Khanal, Bangladesh Institute of Health Sciences

Comments to the Author:

1. Abstract: The objective is not clear. Is it the Rural febrile Illness project or different? Please clarify the objectives

Response:

We thank the reviewer for pointing this out. We have clarified the survey objectives in the abstract as suggested:

“Addressing this knowledge gap, the South and Southeast Asia Community-based Trials Network (SEACTN) will undertake a survey that aims to determine the prevalence of a wide range of non-communicable and communicable diseases, as one of the key initiatives of its first project- the Rural Febrile Illness project (RFI).”

2. Method and analysis.

a. Please specify the non-communicable diseases, if it is already selected.

Response:

The key non-communicable diseases that have been selected are: diabetes, raised blood cholesterol, hypertension, and stroke.

We have revised the “Primary objectives and outcomes” to clarify the point the reviewer raised:

“ Selected non-communicable diseases (e.g. diabetes, raised blood cholesterol, hypertension, stroke) according to self-reported disease history, and/or physical examinations, and/or laboratory tests. ”

b. What is the plan for the family members not in the selected house during the data collection day?

Response:

We thank the reviewer for bringing up this important point. We have now explained this in the section “Survey schedule”, and added a section “Non-responders” in the manuscript.

Survey schedule:

“If household members do not attend the data collection at the appointment time, they will be contacted and encouraged to join the survey at another time during the day, while the research team are in the village.”

Non-responders

“Household members who fail to attend the survey (who are absent, refuse participation, or for other reasons) will have their demographic characteristics (e.g., sex, age) summarized from the information obtained from the household head in the household section of the questionnaire, to determine whether systematic differences exist between responders and non-responders.”

3. Discussion: Are there any possibilities of limitation apart from the selected location?

Response:

We appreciate the reviewer raising this question. We have now added the potential limitations of the study as described below in “Discussion”.

“The cross-sectional design and observational nature of the study is at risk of temporal bias, selection bias, and participation bias. However, to minimize the bias as much as possible, we designed the survey with appropriate evaluation methods, and adequate training and supporting material will be well prepared for the field work.”

Reviewer 2

Dr. Michael von Fricken , George Mason University

Comments to the Author:

1. In your intro you cite the 2019 global burden of disease study when describing the absence of survey data for Cambodia, Myanmar, and Laos, but that doesn't align with your study sites – where you are looking at Bangladesh, Cambodia, and Thailand. Please update, remove, or provide correct context. If you are using GBD I believe there are more recent publications of GBD study as well.

Response:

We thank the reviewer for pointing out the misalignment of the listed countries and the study countries. We have removed this description.

We appreciate the reviewer's suggestion on referencing more recent GBD publications. After careful consideration, although we have identified more recent publications of GBD, the root of these publications are the GBD 2019 study, which is the most recent systematic modeling study on burden of disease globally and have therefore decided to retain our original reference.

2. For your power analysis, shouldn't your target sample size been set on the underlying burden of disease by country? I would suspect the prevalence of some of these diseases would be very different in Thailand vs Bangladesh vs Cambodia – is there any risk of study being underpowered in any location by adopting a one sample size (~1600 per country) fits all approach?

Response:

We agree with the reviewer that the prevalence of disease of each study country should be considered to determine the sample size to ensure adequate power for prevalence estimates. When calculating the sample size, we compiled the estimated prevalence of the targeted diseases from all study countries, and applied the highest value in the calculation to obtain the most conservative sample size estimate. Therefore, the current sample size is designed with sufficient power, as described in the protocol, to detect the prevalence of the target diseases across all sites.

To avoid confusion, we have clarified the section “Sample size”:

“The minimal sample size required was calculated based on the estimated prevalences of key indicators overall and within each population age group (Table 1). Prevalence estimates, population proportions within each age group, and average household size, were derived from previously conducted national health surveys and published studies, preferably from the study country, or from countries with a similar epidemiological context (19–29). When the estimated prevalence of an indicator varied across study country, the highest prevalence estimates were applied.”

3. Please provide more detail on your PCR targets for malaria in your analysis of blood samples section.

Response:

Thanks for the suggestion. The paragraph below have been modified to include the information in section “ Analysis of blood samples”.

“Malaria quantitative polymerase chain reaction (qPCR) will be performed for all participants to identify individuals with malaria parasites. The PCR methods typically have a detection level in the range of 100-1000 parasites per milliliter. Genus-specific qPCR targeting Plasmodium 18S rRNA genes will first be performed to screen for positive individuals, and then species-specific assays will be performed to differentiate the parasite species (Plasmodium falciparum, Plasmodium vivax, Plasmodium malariae, Plasmodium ovale, and Plasmodium knowlesi).”

4. Same for serology tests – are you running ELISAs? IFA? Western blot? MagPIX? A little more granularity here would help flesh out this published protocol.

Response:

Luminex xMAP Intelliflex® System will be used for the serology testing. The detail is included in the manuscript now.

“Multiplex serology tests based on Luminex xMAP Intelliflex® System will be developed and validated for measures of IgG and immunoglobulin M (IgM) of common pathogens causing fever;...”

5. Within your results feedback step – what type of delays are you anticipating between collection event and test results for all HBV, HCV, etc. please include an estimate if available

Response:

Thank you for raising this point. We have now incorporated details about possible delays in the “Results feedback section”.

“It is anticipated that there will be a delay of approximately six months to one-year from sample collection to results feedback. The timeframe encompasses the completion of data collection at the entire site, transportation of samples to Bangkok, sample management, analysis, and the subsequent return of results.”

VERSION 2 – REVIEW

REVIEWER	von Fricken, Michael George Mason University
REVIEW RETURNED	05-Mar-2024

GENERAL COMMENTS	Thank you for addressing all comments.
--